# Synthesis and Characterization of 3-DOM IrO_2_ Electrocatalysts Templated by PMMA for Oxygen Evolution Reaction

**DOI:** 10.3390/polym11040629

**Published:** 2019-04-04

**Authors:** Fei Liu, Xuechu Sun, Xiu Chen, Cuicui Li, Jun Yu, Haolin Tang

**Affiliations:** State Key Laboratory of Advanced Technology for Materials Synthesis and Processing, Wuhan University of Technology, Wuhan 430070, China; lf-news@whut.edu.cn (F.L.); sxc19920506@whut.edu.cn (X.S.); chenxiu@whut.edu.cn (X.C.); Lcc1048556726@163.com (C.L.)

**Keywords:** water electrolysis, OER, PMMA template, IrO_2_ catalyst

## Abstract

Three-dimensional ordered macroporous (3-DOM) IrO_2_ material was prepared using PMMA as a template and ammonia as a chelator. These 3-DOM IrO_2_ honeycomb arrays showed a large surface area and ordered macropores (155 nm in diameter) cross-linked by secondary mesopores. Internal structures of 3-DOM IrO_2_ material were observed microscopically through these secondary pores. According to the X-ray diffraction (XRD) and X-ray photoelectron spectroscopy (XPS) spectra, 3-DOM IrO_2_ has a rutile crystal structure and is mainly composed of iridium dioxide. In acidic electrolytes, the overpotential of 3-DOM IrO_2_ material at 0.5 mV cm^−2^ was only 0.22 V. Accelerated durability tests demonstrated excellent durability of 3-DOM IrO_2_ as an oxygen evolution reaction (OER) catalyst.

## 1. Introduction

Hydrogen, as an environmentally friendly energy carrier with high thermal value, has many advantages, which solves the energy problems and environmental issues associated with other energy resources [1,2,3]. The preparation of hydrogen is mainly based on fossil fuel cracking and water electrolysis. Solid polymer electrolyte (SPE) water electrolysis is considered to be one of the most promising methods for hydrogen production because of its high energy conversion efficiency and power density. Hydrogen production systems are safe and clean. Products of this process are just oxygen and hydrogen, both of which possess high purity [4]. However, some problems associated with SPE water electrolysis still exist, including high anodic overpotential and the high cost of the system [4,5].

Despite numerous research studies in this area, high anodic overpotential remains a great challenge for oxygen evolution reaction (OER). Additionally, the stability of electrocatalysts is another factor restricting widespread SPE applications. The majority of metals and their oxides cannot be stabilized under high anode potential and strong acidic conditions; thus, choices for anode catalysts are very limited. Platinum metals have good catalytic performance because of their excellent adsorption affinity to hydrogen and excellent catalytic properties. Thus, Pt has been widely used in SPE water electrolysis [6,7], fuel cells [8,9,10,11], and hydrogen storage [12,13]. However, Pt-based catalysts are not suitable for OER because of the high overpotential of Pt as it forms platinum oxides, which result in the poor conductivity of Pt [14]. Fortunately, Ir and Ru have good catalytic OER activity. Their oxides also have good conductivity and are stable in acidic environments. Interfaces of Ir and Ru oxides are also very reactive.

To increase the stability of the catalyst at the anode during SPE water electrolysis, a lot of work has been performed especially involving dimensionally stable anodes (DSA). Durable material to load on IrO_2_ and RuO_2_ were selected, most of which were oxides of transition metals. Although these materials have excellent chemical stability, their conductivity is poor, which makes OER electron transfer dependent on IrO_2_ and RuO_2_. The result does not reduce the amount of precious metals, and only when the content of IrO_2_ in the IrO_2_-ZrO_2_ solid solution reaches 80% does the catalyst demonstrate good performance [15]. Other solid oxides forming catalysts with high levels of IrO_2_ and/or RuO_2_ are TiO_2_ [16,17,18,19], SiC [20], SnO_2_ [21,22,23,24], Ta_2_O_5_ [25,26], and Nb_2_O_5_ [27], etc. Research for a variety of other support materials is still underway.

To increase the catalytic activity during SPE water electrolysis, the activity of precious metal catalysts needs to be improved, for example by studying and analyzing its surface area and micropore structure [28,29,30,31]. Recently, Li et al. [30] synthesized a nanoporous IrO_2_ catalyst based on a template-free ammonia complex. Unlike the Adams fusion method, ammonia was used as a chelating agent to replace Cl in H_2_IrCl_4_. This IrO_2_ catalyst demonstrated the highest specific surface area ever reported (363.3 m^2^ g^−1^) as well as outstanding performance. Ortel et al. [31] synthesized ordered mesoporous IrO_2_ film templated by PEO–PB–PEO block copolymers. The films possess nanocrystalline walls and ordered pores, which enhance the durability and activity of OER.

Three-dimensional ordered macroporous (3-DOM) materials are novel molecular sieves with uniform and ordered pores and well-controlled pore structures. The ordered porosity of the 3-DOM material is very conducive to an even distribution of the electrolyte between the electrode, which improves the mass transfer of ions to the electrode surface [32]. This can increase the number of reactive sites and improve the catalytic activity of materials. The interconnected structure of 3-DOM materials can increase conductivity [33], which makes them suitable as electrochemical catalysts. 3-DOM materials were successfully used in a lithium-ion battery and fuel cell [32,34]. However, few 3-DOM oxide materials have been reported for OER electrocatalysts. Hu et al. first synthesized 3-DOM IrO_2_ using a SiO_2_ microsphere template [35,36], in which 3-DOM IrO_2_ exhibited an ordered honeycomb array of macropores. The overpotential of this OER catalyst was only 0.22 V at 0.5 mA cm^−2^. However, organic templates have not been reported for the preparation of 3-DOM IrO_2_. Herein, we synthesized 3-DOM IrO_2_ using a PMMA template. Compared to the SiO_2_ template, PMMA templates can be removed by annealing, which makes the preparation process very simple, thereby reducing its cost. However, the decomposition temperature of the PMMA template is under 300 °C, which is too low for iridium chloride to form iridium oxide. Because the formation of 3-DOM structures is difficult under such conditions, we used ammonia as a chelator, which reacts with iridium chloride to form ammonium chloride and iridium hydroxide. Ammonium chloride is volatile and easily decomposed. Iridium hydroxide is easily dehydrated to produce iridium oxide under 300 °C, which can protect the 3-DOM structure against collapse. 3-DOM IrO_2_ was characterized by SEM, Brunauer-Emmett-Teller (BET), X-ray diffraction (XRD), and X-ray photoelectron spectroscopy (XPS). Ordered macropores were clearly seen with a significant decrease in overpotential.

## 2. Experimental

### 2.1. Preparation of PMMA Templates

Conventional emulsifier-free emulsion polymerization was used to prepare polymethyl methacrylate (PMMA) templates [37]. First, 260 mL of deionized water was placed into a 500 mL round-bottomed flask and heated to 70 °C inside a heat-collecting thermostatic magnetic stirrer. After nitrogen was passed through the flask for 15 min, potassium persulfate (KPS, Sinopharm Chemical Reagent Co., Ltd., Shanghai, China) and methyl methacrylate (MMA, Sinopharm Chemical Reagent Co., Ltd., Shanghai, China) were added in a ratio of 1:118. The whole mixture was stirred for 1 h under nitrogen. After that, the PMMA emulsion was centrifuged and washed three times with deionized water, following which PMMA microspheres were dispersed ultrasonically in deionized water. A certain amount of PMMA emulsion was placed into a 25 mL beaker, which was then covered by tin foil with holes. PMMA microspheres were settled down and dried at room temperature. Bright fluorescence on the surface indicated that the PMMA microspheres were hexagonally close-packed.

### 2.2. Preparation of 3-DOM IrO_2_

IrCl_4_·2H_2_O (Shanxi Kaida Chemical Engineering Co., Ltd., Shanxi, China) was added to H_2_O and isopropyl alcohol (IPA, Sinopharm Chemical Reagent Co., Ltd., Shanghai, China) (H_2_O: IPA = 3:1). Then, 4 mL of IrCl_4_·2H_2_O 0.03 mol L^−1^ was added to PMMA templates until it was immersed. After 30 min, the excess solution was removed by a dropper. PMMA templates were put into the oven at 70 °C. After drying, the above steps were repeated three times. Finally, the templates were heated to 450 °C in air at a rate of 5 °C/min in a tube furnace for 4 h. The final product was a black powder (marked as non-ammonia IrO_2_). Ammonia (Sinopharm Chemical Reagent Co., Ltd., Shanghai, China) (IrCl_4_·nH_2_O: NH_4_·H_2_O = 1:4) was then added to the above solution that contains isopropyl alcohol, H_2_O, and IrCl_4_·2H_2_O. 3-DOM IrO_2_ was prepared by repeated preparation of non-ammonia IrO_2_. Colloidal IrO_2_ was prepared by the colloidal method [35].

### 2.3. Physical Characterization

Scanning electron microscopy (SEM) was performed using a Hitachi S-4800 microscope operated at 8 kV. X-ray photoelectron spectroscopy (XPS) spectra were recorded by an ESCALAB250Xi spectrometer using Mg Kα as an X-ray source. X-ray diffraction (XRD) spectra were recorded by Bruker D8 Advance using Cu–Kα radiation. Particle sizes were measured by the dynamic light scattering (DLS) technique. Specific surface areas and pore distribution were obtained by implementing Brunauer-Emmett-Teller (BET) and Barrett-Joyner-Halenda (BJH) methods, respectively, using N_2_ gas absorption/desorption data collected by Micromeritics ASAP 2020 (BELSORP-max).

### 2.4. Electrochemical Characterization

Electrochemical properties were tested using a three-electrode system at room temperature. The electrolyte was 0.5 mol L^−1^ H_2_SO_4_ solution. The counter electrode was platinum foil, and a saturated calomel electrode (SCE, 0.242 V) was a reference electrode. The working electrode was a glassy carbon (GC) 5 mm in diameter. The catalyst ink was prepared using 5 mg of testing material, 980 µL of isopropanol, and 20 µL of 5% Nafion. Both catalysts had a load of 106 µg cm^−2^. Cyclic voltammograms (CV) were measured between 0 V and 1.4 V at a 50 mV s^−1^ scan rate. Steady-state polarization curves for OER were obtained at a 5 mV s^−1^ scan rate. Before all tests, the electrode was activated by performing 20 cycles of the CV curve at the scanning rate (500 mV s^−1^). All electrochemical tests were performed using the CHI660 workstation.

## 3. Results and Discussion

### 3.1. Physical Characterization

Since PMMA microspheres serve as a synthesis template, their particle size distribution is an important physical property. Analysis of the particle size distribution of PMMA by DLS demonstrates that the average diameter of PMMA microspheres is ~220 nm (Figure 1a). Polydispersity of PMMA microspheres is 1.7%, which is indicative of good uniformity. A single particle size of PMMA microspheres is an important reason for forming a regular colloidal crystal template. Also, sizes of PMMA microspheres are controlled by the quantity of the initiator and MMA. The PMMA template consists of hexagonally close-packed PMMA microspheres (Figure 1b), which have a uniform distribution of particle size (the diameter of the particles is 220 nm). An obvious layered structure can be clearly observed in Figure 1b. All layers have the same close-packed hexagonal arrangement of the microspheres. Several point defects are observed as well as some microsphere misalignments (markings in Figure 1b). These defects have no impact on the template ordered macroporous material formed because of this ordered regular template. The PMMA template is covered by the precursor, as observed in Figure 1d. After immersion, the PMMA template retains the same close-packed hexagonal arrangement of the microspheres. The PMMA colloidal crystal template exhibits fluorescence (Figure 1c) under natural sunlight, which is a typical Bragg behavior of ordered microspheres.

3-DOM IrO_2_ exhibits a three-dimensional ordered honeycomb macroporous structure (Figure 2). Most of the macropores are 155 nm in diameter, which is ~30% smaller than the diameters of PMMA microspheres in the template, probably because of the carbonization and dehydration of the organic templates during annealing. PMMA contains many hydrogen and oxygen elements, and dehydration can greatly reduce the microsphere size as well as carbonize the PMMA template. The honeycomb pore structure can be seen in the high-magnification SEM image. In addition, small pores in the walls of the large pores, which can connect to the surrounding large pore structures, can also be clearly observed (Figure 2b). These pore structures are formed because when accumulated hexagonally close-packed PMMA microspheres are in contact with each other, the precursor solution cannot pass through the points of contact of the PMMA spheres during impregnation. After the removal of the PMMA template, these contact points form small pores. The solid structure of the 3-DOM IrO_2_ can be seen through these small pores. These pores are favorable for the even distribution of the electrolyte between the electrodes, which will improve the mass transfer of ions to the surface of the electrodes. This can also increase reactive sites and improve the catalytic activity of the material. Additionally, such interconnected structures can increase the overall conductivity. A few pores can be observed in non-ammonia IrO_2_ (Figure 2c). These pores are broken and disordered probably because the decomposition temperature of IrCl_4_ is higher than that of the organic template. Before IrCl_4_ decomposes, the decomposition of the organic template causes the pore structure to collapse (Figure 2d).

The specific surface area of the OER catalyst is an important physical property. Because of its microporous structure, it has a large specific surface area. Figure 3a exhibits N_2_ adsorption/desorption isotherms for 3-DOM IrO_2_ and non-ammonia IrO_2_. According to the International Union of Pure and Applied Chemistry (IUPAC) classification [38], the curve of 3-DOM IrO_2_ belongs to the type IV with H3-type hysteresis, which is usually associated with highly consistent mesopore structures. Non-ammonia IrO_2_ also shows the type IV with H3-type hysteresis. However, the hysteresis loop is smaller than that in 3-DOM IrO_2_, showing that 3-DOM IrO_2_ has more mesopores. According to the BJH pore-size distribution analysis (Figure 3b), the pore sizes of both materials are concentrated below 20 nm. Diameters of 3-DOM IrO_2_ mesopores are ~1.2 nm, while diameters of non-ammonia IrO_2_ mesopores are ~2.1 nm. The BJH surface areas of the mesopores of 3-DOM IrO_2_ and non-ammonia IrO_2_ are 34.5 and 23.8 m^2^ g^−1^, respectively, indicating that 3-DOM IrO_2_ has more mesopores. The BET surface areas of 3-DOM IrO_2_ and non-ammonia IrO_2_ are 67.3 and 37.1 m^2^ g^−1^, respectively. The BET surface area of 3-DOM IrO_2_ is close to the mesoporous IrO_2_ structure reported by Ortel et al. [31].

Figure 4 shows the XRD patterns of two kinds of materials. XRD patterns of 3-DOM IrO_2_ showing diffraction peaks corresponding to the rutile IrO_2_ peak at 2θ angles equal to 28°, 34°, 40°, 54°, and 58°, which correspond to the (110), (101), (200), (211), and (220) planes, can be easily seen.

Non-ammonia IrO_2_ pattern does not show rutile IrO_2_ peaks. However, characteristic peaks of the elemental Ir at 2θ of ~40°, 47°, and 69° (which correspond to (111), (200), and (220) planes) could be clearly seen. This indicates that 3-DOM IrO_2_ contained metallic Ir formed because of template carbonization; after PMMA carbonization, internal Ir^4+^ had no access to air. Thus, a small amount of IrO_2_ was reduced to Ir. However, Ir^4+^ in the precursor that has no ammonia is IrCl_4_, and since Cl is less electronegative than O, Ir^4+^ was reduced almost entirely.

XPS analysis of 3-DOM suggests that IrO_2_ is present in the catalyst (Figure 5). No other impurities are observed on the surface (Figure 5a). Characteristic peaks of Ir are clearly seen as well. High-resolution spectra of the Ir4f demonstrates characteristic peaks at 61.5 eV (Figure 5b). This binding energy value agrees with values obtained from nanoporous IrO_2_ synthesized by the template-free ammonia complex [30].

### 3.2. Electrocatalytic Characterization

The electrocatalytic performances of two materials are shown in Figure 6. The peaks observed in the CV curve of colloidal IrO_2_ (Figure 6a) clearly indicate Ir transformation (Ir^III^/Ir^IV^ and Ir^IV^/Ir^IV^), accompanied by protonation/deprotonation. The CV curve of 3-DOM IrO_2_ demonstrated a distinct redox peak at a low potential, which was related to the elemental Ir in 3-DOM IrO_2_. Ir, similar to Pt and Ru, can absorb hydrogen in the hydrogen underpotential deposition region [39,40]. 3-DOM IrO_2_ showed significantly larger current density in comparison to colloidal IrO_2_. Current responses consist of two effects, the redox pseudo-capacitance and the double-layer capacitance, both of which are related to the electroactive surface areas (ESAs).

The linear sweep voltammetry (LSV) of two catalysts demonstrating their OER activities are shown in Figure 6b. 3-DOM IrO_2_ demonstrates superior OER activity compared to colloidal IrO_2_. Compared to the thermodynamic minimum potential of OER, the overpotential of 3-DOM IrO_2_ is 0.22 V at 0.5 mA cm^−2^, which outperforms colloidal IrO_2_ (0.25 V) and is the same as the 3-DOM IrO_2_ material synthesized by the silica colloidal crystal template [35] and the mesoporous IrO_2_ templated by PEO–PB–PEO copolymers [31]. However, the preparation of PMMA microspheres was easy as 3-DOM IrO_2_ prepared by the PMMA template did not require acid solution treatment, thereby reducing the cost of preparation. Figure 6b suggests that 3-DOM IrO_2_ (106 µg cm^−2^) has the same catalytic activity as colloidal IrO_2_ (265 µg cm^−2^), while the OER activity of 3-DOM IrO_2_ is outstanding compared to that of colloidal IrO_2_, because of their different ESAs (Figure 6a). The rutile IrO_2_ shows lower O_2_-evolving overpotential than the metal Ir [41]; therefore, the metal Ir is not the main reason for the increase of catalytic activity. Catalytic data were normalized relative to the corresponding CV charges (inset of Figure 6c). The curves represent intrinsic activities of the catalysts, which are almost identical (Figure 6c). Thus, intrinsic OER activities are identical, and the difference in porosity does not affect catalytic performance. However, 3-DOM IrO_2_ shows superior catalytic performance under the same mass condition. The slope of the Tafel curve for 3-DOM IrO_2_ is 57.9 mV dec^−1^ (Figure 6d), which is lower than that of colloidal IrO_2_ (which is equal to 62.2 mV dec^−1^). The Tafel slope is similar to that of nanoporous IrO_2_ and within the normal range of conventional IrO_2_ [30]. This also confirms the robust catalytic activity of 3-DOM IrO_2_, indicating a faster rate for the OER.

Figure 7 shows current change as a function of time. From these curves, the OER stabilities of both catalysts in acidic environments can be obtained. Current density dropped initially for both catalysts. After 3000 s, the decline of the current density slowed down. However, 3-DOM IrO_2_ still showed higher current density than that of colloidal IrO_2_. Slopes of the fitting curves in the range of 15,000–20,000 s (insert in Figure 7) are −2.09 × 10^−4^ and −2.37 × 10^−4^, respectively. Thus, the current change of colloidal IrO_2_ deceases faster and 3-DOM IrO_2_ exhibits superior stability than that of colloidal IrO_2_.

## 4. Conclusions

Synthesis of 3-DOM IrO_2_ using a PMMA template and ammonia sol has been performed. The issue of the organic template having difficulty in maintaining the 3-DOM IrO_2_ structure has been solved by the ammonia sol. The sol consists of iridium chloride, isopropanol, and ammonia. IrCl_4_ and ammonia form a stable complex solution, in which NH_4_^+^ and Cl^−^ form ammonium chloride, and ammonium chloride can easily be removed. The prepared 3-DOM IrO_2_ material has a honeycomb array macroporous structure with an average pore diameter of 155 nm and a large surface area. Many mesoscale pores are observed on the wall of the macropores. 3-DOM IrO_2_ demonstrates superior OER activity compared to colloidal IrO_2_, and the overpotential of 3-DOM IrO_2_ is lower than that of colloidal IrO_2_ by about 30 mV at 0.5 mA cm^−2^. In the same catalytic activity, the loading of 3-DOM IrO_2_ is much lower than that of colloidal IrO_2_. Our novel method to synthesize the 3-DOM IrO_2_ catalyst can reduce the anode overpotential of water electrolysis. The results suggest that the 3-DOM IrO_2_ catalyst is a promising material for OER.

## Figures and Tables

**Figure 1 polymers-11-00629-f001:**
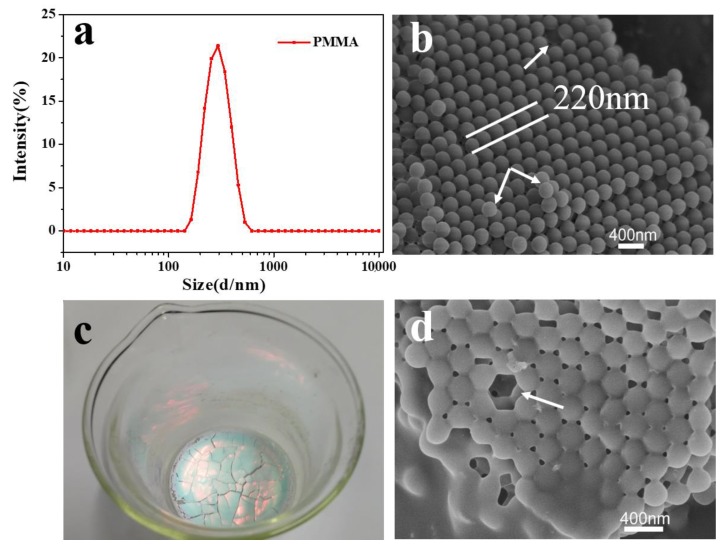
(**a**) Dynamic light scattering (DLS) particle size distribution of PMMA emulsion; (**b**) SEM surface image of PMMA template; (**c**) natural fluorescence of PMMA emulsion under natural sunlight; (**d**) SEM surface image of PMMA template covered by the precursor.

**Figure 2 polymers-11-00629-f002:**
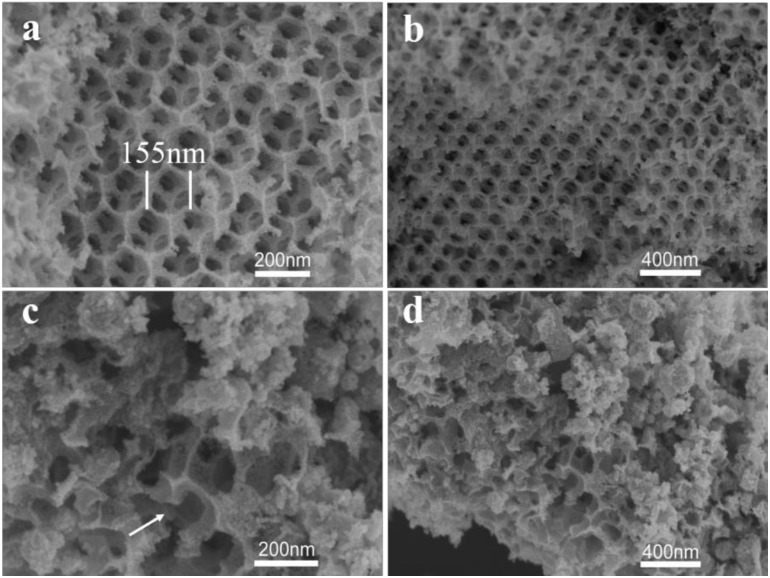
(**a**) High- and (**b**) low-magnification SEM images of three-dimensional ordered macroporous (3-DOM) IrO_2_ honeycomb structures; (**c**) high- and (**d**) low-magnification SEM images of non-ammonia IrO_2_ structures.

**Figure 3 polymers-11-00629-f003:**
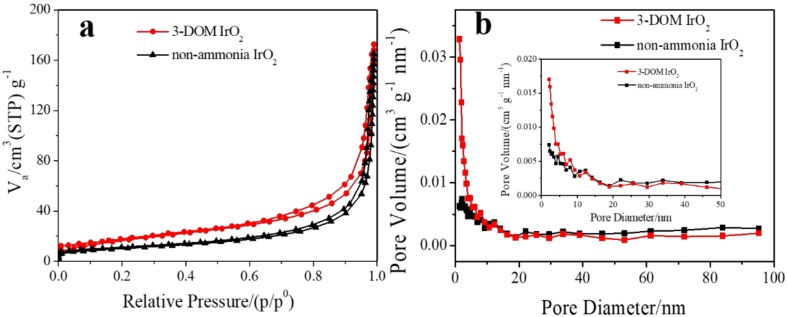
(**a**) N_2_ adsorption/desorption isotherms; (**b**) BJH pore-size distribution for 3-DOM IrO_2_ and non-ammonia IrO_2_.

**Figure 4 polymers-11-00629-f004:**
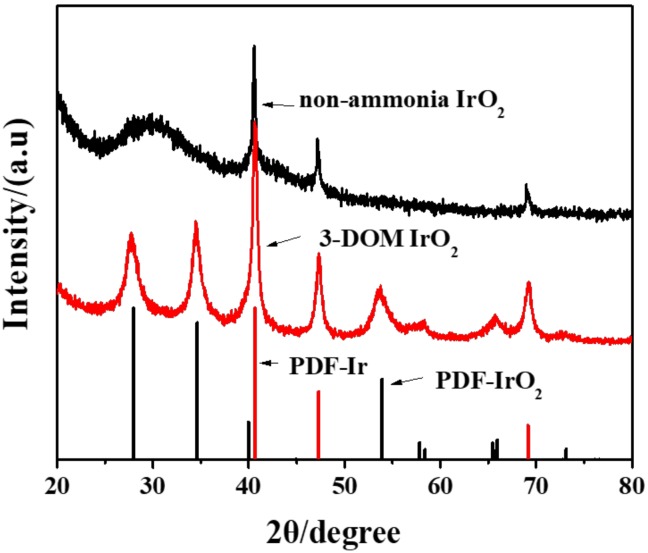
XRD patterns of 3-DOM IrO_2_ and colloidal IrO_2_.

**Figure 5 polymers-11-00629-f005:**
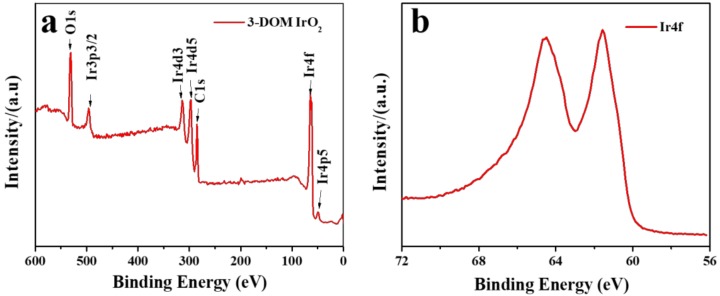
(**a**) X-ray photoelectron spectroscopy (XPS) spectra of 3-DOM IrO_2_; (**b**) high-resolution spectra of the Ir4f peak.

**Figure 6 polymers-11-00629-f006:**
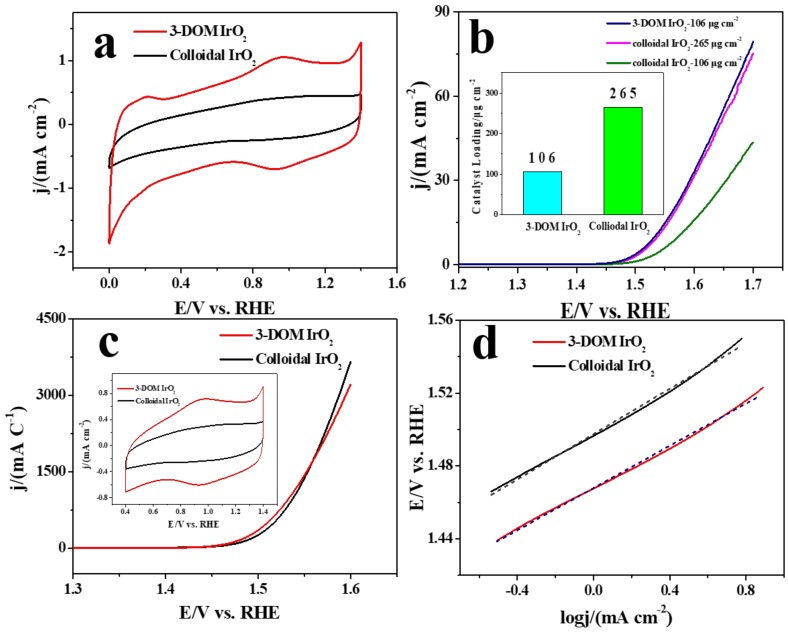
(**a**) Cyclic voltammetry (CV) curves; (**b**) steady-state polarization curves with 106 µg cm^−2^ loadings; (**c**) the current density is normalized to CV; (**d**) Tafel curves for 3-DOM IrO_2_ and colloidal IrO_2_.

**Figure 7 polymers-11-00629-f007:**
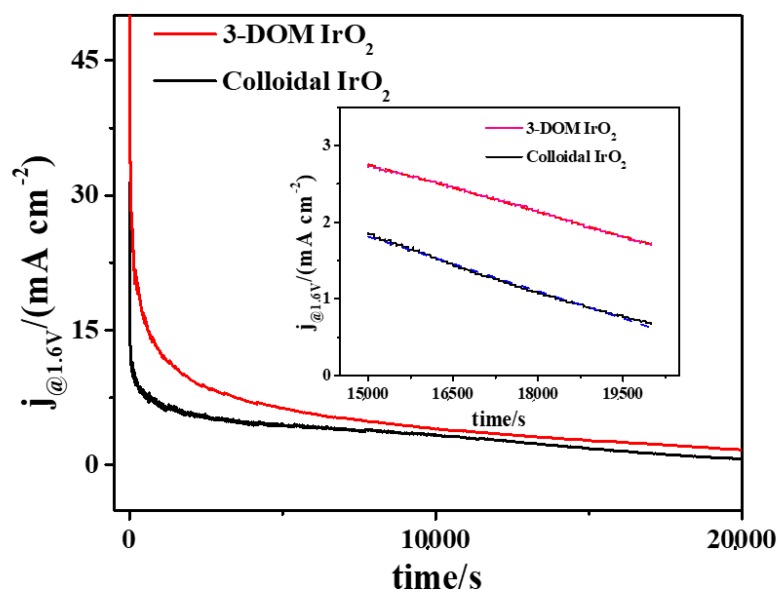
Time evolution currents for two IrO_2_ materials under a constant potential (1.6 V) in 0.5 mol L^−1^ H_2_SO_4_ solution.

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
