# Peer review of "Synthesis and Characterization of 3-DOM IrO2 Electrocatalysts Templated by PMMA for Oxygen Evolution Reaction"

_polymers, 2019, doi:10.3390/polym11040629_

Round 1
Reviewer 1 Report
In this paper, authors discuss the performance of 3-DOM IrO2 sythesized via organic PMMA templated process. Both physical and electrochemical characterization results were presented. I do not recommended the article for publication in its current form. Major revisions are required before considering the manuscript for publication.
The level of English is not adequate. Here are a few examples
1. Page 1, line 11 - "macropores" not "macrospores"
2. Page 3, line 62 delete the word "prepared"
3. Page 3, line 70 "is volatile" not "are volatile"
The whole manuscript need to be properly revised so that the ideas and discussion points come across clearly to the reader.
In Figure 3a, the authors claim that based on the hysteresis of the loops , 3-DOM IrO2 has more mesopores. However, this visual comparison does not conclusively convey the point. The authors need to include the BJH plot that is just focused between 2 and 50 nm. Also, the relevant quantitative information needs to be included.
The OER activity of the 3 DOM IrO2 (based on the overpotential at 0.5 mA/cm2 ) is marginally better than colloidal IrO2 and similar to the catalyst in literature (Ref 29, Ortel et al.). The authors should explain the benefits of using organic PMMA templated catalyst (for example, cost vs performance benefit, durability etc.)
As the OER activity of 3 DOM IrO2 is similar to the ones reported in literature, the only advantage seems to be the durability. While the authors show some durability performance in Figure 7, the plot stops at 10000 seconds which is way too short of a duration to represent real life operation. The durability study needs to be extended for a few days. Also, it should be compared with the durability of similar catalysts reported in literature.
Author Response
Response to Reviewer #1:
We first take this opportunity to give our thanks to you for the effort made on this manuscript. We have carefully considered the comments arisen from you and had made according changes as highlighted with red colors in the revised manuscript.
1.The level of English is not adequate. Here are a few examples
Page 1, line 11 - "macropores" not "macrospores"
Page 3, line 62 delete the word "prepared"
Page 3, line 70 "is volatile" not "are volatile"
The whole manuscript need to be properly revised so that the ideas and discussion points come across clearly to the reader
Response: We thank the reviewer for the comment and it has been corrected in Page 1, line 11, Page 3, line 62 and Page 3, line 70 in the revised manuscript as highlighted with red colors. We are sorry for the grammatical errors, typos, and less-understandable language. We have carefully checked and revised the manuscript and the according changes have been made throughout the manuscript as highlighted with red colors in the revised version.
2. In Figure 3a, the authors claim that based on the hysteresis of the loops, 3-DOM IrO2 has more mesopores. However, this visual comparison does not conclusively convey the point. The authors need to include the BJH plot that is just focused between 2 and 50 nm. Also, the relevant quantitative information needs to be included.
Response: We thank the reviewer for the suggestion. The BJH plot that is just focused between 2 and 50 nm has been added, the pore sizes of both materials are concentrated below 20nm. BJH surface areas of the mesoporous of 3-DOM IrO2 and non-ammonia IrO2 are 34.5 and 23.8 m2 g-1, these can indicate that 3-DOM IrO2 has more mesopores. Such description was added in page 5 in the revised manuscript as highlighted with red colors.
3.The OER activity of the 3 DOM IrO2 (based on the overpotential at 0.5 mA/cm2) is marginally better than colloidal IrO2 and similar to the catalyst in literature (Ref 29, Ortel et al.). The authors should explain the benefits of using organic PMMA templated catalyst (for example, cost vs performance benefit, durability etc.)
Response: We thank the reviewer for the suggestion. Comparing to other templates, the preparation of PMMA microspheres is easy, 3-DOM IrO2 prepared by PMMA template does not need acid solution treatment, So the cost of preparation will be reduced. To compare the catalytic activity of the two catalysts, the polarization curve of colloidal IrO2 at 265 µg cm-2 has been measured. Figure. 6b suggests that 3-DOM IrO2 (106 µg cm-2) had the same catalytic activity as colloidal IrO2 (265 µg cm-2). these indicate that OER activity of 3-DOM IrO2 is superior than of colloidal IrO2. We have accordingly modified the discussion in page 7 in the revised manuscript as highlighted with red colors.
3.As the OER activity of 3-DOM IrO2 is similar to the ones reported in literature, the only advantage seems to be the durability. While the authors show some durability performance in Figure 7, the plot stops at 10000 seconds which is way too short of a duration to represent real life operation. The durability study needs to be extended for a few days. Also, it should be compared with the durability of similar catalysts reported in literature.
Response: We sincerely thank the reviewer for the comment. the plot stops at 20000 seconds which has been added. Current density dropped initially for both catalysts. After 3000 s, the decline of current density slowed down. However, 3-DOM IrO2 still show higher current density than that of colloidal IrO2. Slopes of the fitting curves in the 15000-20000 s time internal (Insert in Fig. 7) are -2.09·10-4 and -2.37·10-4. This suggests that current change of colloidal IrO2 deceases faster. We have accordingly modified the discussion in page 7 in the revised manuscript as highlighted with red colors.

Reviewer 2 Report
The authors describe a method for the synthesis of 3-DOM material templated by PMMA. The Ir-based material obtained has shown better performance for OER than the colloidal Ir-based catalyst. The manuscript is quite well written and easy to follow and understand. However, there are some points that must be adressed:
In point 2 (Experimental) it lacks the materials description with all reagents and solvents employed. A person not used to scientific literature wolud have some problems for the fast identification of PMMA and KPS compounds, for example.
The point 2.2 of preparation of 3-DOM-IrO2 should be improved in the explanation and details regarding the synthesis of the material. In line 87 there is a point missing; In the same line it should be good indicate the approximate volume added to the PMMA template. Then in line 88, solution is decanted or removed? If removed, is by centrifugation? If decanted (PMMA spheres) how long for decantation time?
Conditions for the annealing in terms of ramp (ºC/min to reach annealing T) and atmosphere should also be detailed.
In line 92 "to the above solution", please especify which is the solution; also, I am not able to understand the meaning of the next sentence.
Section 3: Results and discussion. Line 120: particle diameter is referred twice.
In line 123, the authors refer to defects: it is posible to know the defects percentage? and to remove such defects for example with solvent washing?
Figure 1 and 2: it would be better to choose another color than red, especially for people printing in grey scale.
Novelty of the manuscript and relevance is not supported by the references selected. It would be good to look for some more recent bibliography.
Author Response
Response to Reviewer #2:
We first take this opportunity to give our thanks to you for the effort made on this manuscript. We have carefully considered the comments arisen from you and had made according changes as highlighted with red colors in the revised manuscript.
1.In point 2 (Experimental) it lacks the materials description with all reagents and solvents employed. A person not used to scientific literature would have some problems for the fast identification of PMMA and KPS compounds, for example.
Response: We apologize for the unclear description. We have added the full name of these chemicals, such as: polymethyl methacrylate (PMMA), potassium persulfate (KPS) and methyl methacrylate (MMA). We have accordingly modified in page 2 and page 3 in the manuscript as highlighted with red colors.
2.The point 2.2 of preparation of 3-DOM IrO2 should be improved in the explanation and details regarding the synthesis of the material. In line 87 there is a point missing; In the same line it should be good indicate the approximate volume added to the PMMA template. Then in line 88, solution is decanted or removed? If removed, is by centrifugation? If decanted (PMMA spheres) how long for decantation time?
Response: We sincerely thank the reviewer for the comment. We have explained details regarding the synthesis of the material, such as: the concentration of IrCl4·2H2O solution is about 0.03 mol L-1, the amount of solution added is 4mL, the solution is removed by dropper. All modifications have been marked red in page 2 and page 3 in the manuscript.
3.In line 92 "to the above solution", please especify which is the solution; also, I am not able to understand the meaning of the next sentence.
Response: In line 99 “to the above solution”, this solution is IrCl4·2H2O solution which is about 0.03 mol L-1. About next sentence, what I want to express is that 3-DOM IrO2 was prepared by repeated preparation of non-ammonia IrO2. We are sorry for the unclear description and this has been corrected in page 2 in the revised manuscript as highlighted with red colors.
4.Conditions for the annealing in terms of ramp (ºC/min to reach annealing T) and atmosphere should also be detailed
Response: We very much appreciate the comment arisen from the reviewer. The templates were heated to 450 ℃ in air at a rate of 5 ℃/min in tube furnace for 4h. We have modified in page 3 the manuscript as highlighted with red colors.
5.Section 3: Results and discussion. Line 120: particle diameter is referred twice.
Response: We very much appreciate the comment arisen from the reviewer. Particle diameter is referred twice, because they are measured in different ways. particle diameter is measured by DLS in the first time. particle diameter is measured by SEM in the second time.
6.In line 123, the authors refer to defects: it is posible to know the defects percentage? and to remove such defects for example with solvent washing?
Response: We thank the reviewer for the comment. Defects are formed during deposition and the corresponding percentage cannot be measured, and these defects do not affect the overall regularity of the template.
7.Figure 1 and 2: it would be better to choose another color than red, especially for people printing in grey scale.
Response: We very much appreciate the comment arisen from the reviewer. the red label has been replaced by white. I hope this is suitable for printing in grey scale.
8. Novelty of the manuscript and relevance is not supported by the references selected. It would be good to look for some more recent bibliography.
Response: We thank the reviewer for the suggestion, some recent bibliography has been quoted,
References:
4. Sharma, S.; Ghoshal, S.K. Hydrogen the future transportation fuel: From production to applications. Renewable & Sustainable Energy Reviews 2015, 43, 1151-1158.
5. Schmidt, O.; Gambhir, A.; Staffell, I.; Hawkes, A.; Nelson, J.; Few, S. Future cost and performance of water electrolysis: An expert elicitation study. International Journal of Hydrogen Energy 2017, 42, 30470-30492
6. Sapountzi, F.M.; Gracia, J.M.; Weststrate, C.J.; Fredriksson, H.O.A.; Niemantsverdriet, J.W. Electrocatalysts for the generation of hydrogen, oxygen and synthesis gas. Progress in Energy and Combustion Science 2017, 58, 1-35.
9. Kaewsai, D.; Yeamdee, S.; Supajaroon, S.; Hunsom, M. Orr activity and stability of PtCr/C catalysts in a low temperature/pressure PEM fuel cell: Effect of heat treatment temperature. International Journal of Hydrogen Energy 2018, 43, S0360319918301769.
12. Kishor, R.; Singh, S.B.; Ghoshal, A.K. Role of metal type on mesoporous KIT-6 for hydrogen storage. International Journal of Hydrogen Energy 2018, S0360319918312643
16. Lu, Z.-X.; Shi, Y.; Yan, C.-F.; Guo, C.-Q.; Wang, Z.-D. Investigation on IrO2 supported on hydrogenated TiO2 nanotube array as OER electro-catalyst for water electrolysis. International Journal of Hydrogen Energy 2017, 42, 3572-3578.
19. Oakton, E.; Lebedev, D.; Povia, M.; Abbott, D.F.; Fabbri, E.; Fedorov, A.; Nachtegaal, M.; Copéret, C.; Schmidt, T.J. IrO2-TiO2: A high-surface-area, active, and stable electrocatalyst for the oxygen evolution reaction. ACS Catalysis 2017, 7, 2346-2352.
24. Jiang, G.; Yu, H.; Hao, J.; Chi, J.; Fan, Z.; Yao, D.; Qin, B.; Shao, Z. An effective oxygen electrode based on Ir0.6Sn0.4O2 for pem water electrolyzers. Journal of Energy Chemistry 2019, 39, 23-28
25. Yan, Z.; Zhang, H.; Feng, Z.; Tang, M.; Yuan, X.; Tan, Z. Promotion of in situ tin x interlayer on morphology and electrochemical properties of titanium based IrO2-Ta2O5 coated anode. Journal of Alloys & Compounds 2017, 708, 1081-1088
32. Lei, W.; Wang, F.; Zhu, J.; Xin, Z.; Yi, T.; Xing, W. Synthesis and electrochemical performance of three-dimensional ordered hierarchically porous Li4Ti5O12 for high performance lithium ion batteries. Ceramics International 2017, 44, S0272884217317418.
37. Haddadine, N.; Agoudjil, K.; Abouzeid, K.; Castano, C.E.; Bouslah, N.; Benaboura, A.; Samy El-Shall, M. Optical and physical properties of iridescent photonic crystals obtained by self-assembled polymethyl methacrylate nanospheres within graphene oxide nanoplatelets. Polymers for Advanced Technologies 2018, 29, 244-253
40. Papaderakis, A.; Tsiplakides, D.; Balomenou, S.; Sotiropoulos, S. Probing the hydrogen adsorption affinity of Pt and Ir by surface interrogation scanning electrochemical microscopy (si-secm). Electrochemistry Communications 2017, 83, 77-80

Reviewer 3 Report
Summary and general comment:
Good material design for efficient oxygen evolution reaction is important for hydrogen energy technology. This contribution reports the three-dimensional ordered macroporous IrO2 material as a OER catalyst. The overpotential at 0.5 mV cm-2 is 0.22. A few comments below are listed for the authors as a reference.
Additional Comments:
1. Figure 3b is not mentioned in the manuscript.
2. Why comment that “3-DOM IrO2 and non-ammonia IrO2 show relatively uniform mesopores” in the manuscript? The BJH data show higher microporosity of 3-DOM IrO2 than non-ammonia IrO2.
3. The secondary mesopores are important morphology in the material. It is suggested to analyze the content, surface area and prove volume of mesopore in the material.
4. A good reference survey and a good research introduction are reported. However, it would be better to update the reference. The newest reference cited is 2017. Are there no good researches in 2018?
Author Response
Response to Reviewer #3:
We first take this opportunity to give our thanks to you for the effort made on this manuscript. We have carefully considered the comments arisen from you and had made according changes as highlighted with red colors in the revised manuscript.
1. Figure 3b is not mentioned in the manuscript.
Response: We apologize for not mentioning Figure 3b. it has been mentioned in the manuscript. This was briefly added in page 5 in the revised manuscript as highlighted with red colors.
2.Why comment that “3-DOM IrO2 and non-ammonia IrO2 show relatively uniform mesopores” in the manuscript? The BJH data show higher microporosity of 3-DOM IrO2 than non-ammonia IrO2.
Response: We very much appreciate the comment arisen from the reviewer. What we want to express is that the pore sizes of both materials are concentrated below 20 nm. We have accordingly modified in Page 5 in the manuscript as highlighted with red colors.
3.The secondary mesopores are important morphology in the material. It is suggested to analyze the content, surface area and prove volume of mesopore in the material.
Response: We thank the reviewer for the comment. The BJH plot that is just focused between 2 and 50 nm has been added, the pore sizes of both materials are concentrated below 20nm. BJH surface areas of the mesoporous of 3-DOM IrO2 and non-ammonia IrO2 are 34.5 and 23.8 m2 g-1.
4.A good reference survey and a good research introduction are reported. However, it would be better to update the reference. The newest reference cited is 2017. Are there no good researches in 2018?
Response: We thank the reviewer for the suggestion. We have quoted some new reference.
References:
4. Sharma, S.; Ghoshal, S.K. Hydrogen the future transportation fuel: From production to applications. Renewable & Sustainable Energy Reviews 2015, 43, 1151-1158.
5. Schmidt, O.; Gambhir, A.; Staffell, I.; Hawkes, A.; Nelson, J.; Few, S. Future cost and performance of water electrolysis: An expert elicitation study. International Journal of Hydrogen Energy 2017, 42, 30470-30492
6. Sapountzi, F.M.; Gracia, J.M.; Weststrate, C.J.; Fredriksson, H.O.A.; Niemantsverdriet, J.W. Electrocatalysts for the generation of hydrogen, oxygen and synthesis gas. Progress in Energy and Combustion Science 2017, 58, 1-35.
9. Kaewsai, D.; Yeamdee, S.; Supajaroon, S.; Hunsom, M. Orr activity and stability of PtCr/C catalysts in a low temperature/pressure PEM fuel cell: Effect of heat treatment temperature. International Journal of Hydrogen Energy 2018, 43, S0360319918301769.
12. Kishor, R.; Singh, S.B.; Ghoshal, A.K. Role of metal type on mesoporous KIT-6 for hydrogen storage. International Journal of Hydrogen Energy 2018, S0360319918312643
16. Lu, Z.-X.; Shi, Y.; Yan, C.-F.; Guo, C.-Q.; Wang, Z.-D. Investigation on IrO2 supported on hydrogenated TiO2 nanotube array as OER electro-catalyst for water electrolysis. International Journal of Hydrogen Energy 2017, 42, 3572-3578.
\\\\19. Oakton, E.; Lebedev, D.; Povia, M.; Abbott, D.F.; Fabbri, E.; Fedorov, A.; Nachtegaal, M.; Copéret, C.; Schmidt, T.J. IrO2-TiO2: A high-surface-area, active, and stable electrocatalyst for the oxygen evolution reaction. ACS Catalysis 2017, 7, 2346-2352.
24. Jiang, G.; Yu, H.; Hao, J.; Chi, J.; Fan, Z.; Yao, D.; Qin, B.; Shao, Z. An effective oxygen electrode based on Ir0.6Sn0.4O2 for pem water electrolyzers. Journal of Energy Chemistry 2019, 39, 23-28
25. Yan, Z.; Zhang, H.; Feng, Z.; Tang, M.; Yuan, X.; Tan, Z. Promotion of in situ tin x interlayer on morphology and electrochemical properties of titanium based IrO2-Ta2O5 coated anode. Journal of Alloys & Compounds 2017, 708, 1081-1088
32. Lei, W.; Wang, F.; Zhu, J.; Xin, Z.; Yi, T.; Xing, W. Synthesis and electrochemical performance of three-dimensional ordered hierarchically porous Li4Ti5O12 for high performance lithium ion batteries. Ceramics International 2017, 44, S0272884217317418.
37. Haddadine, N.; Agoudjil, K.; Abouzeid, K.; Castano, C.E.; Bouslah, N.; Benaboura, A.; Samy El-Shall, M. Optical and physical properties of iridescent photonic crystals obtained by self-assembled polymethyl methacrylate nanospheres within graphene oxide nanoplatelets. Polymers for Advanced Technologies 2018, 29, 244-253
40. Papaderakis, A.; Tsiplakides, D.; Balomenou, S.; Sotiropoulos, S. Probing the hydrogen adsorption affinity of Pt and Ir by surface interrogation scanning electrochemical microscopy (si-secm). Electrochemistry Communications 2017, 83, 77-80

Round 2
Reviewer 1 Report
The authors addresses most of the concerns thereby improving the manuscript quality. I recommend the article for publication after addressing these concerns.
The units of catalyst loading figure (mg/cm2) does not match with the units mentioned in the text (ug/cm2). This needs to be corrected.
While the authors improved the quality of english of the manuscript, I still spotted a few grammatical errors in the manuscript. I recommend the authors to revise the manuscript again to correct these errors
Author Response
Response to Reviewer #1:
We first take this opportunity to give our thanks to you for the effort made on this manuscript. We have carefully considered the comments arisen from you and had made according changes as highlighted with red colors in the revised manuscript.
1. The authors addresses most of the concerns thereby improving the manuscript quality. I recommend the article for publication after addressing these concerns.
The units of catalyst loading figure (mg/cm2) does not match with the units mentioned in the text (ug/cm2). This needs to be corrected.
While the authors improved the quality of English of the manuscript, I still spotted a few grammatical errors in the manuscript. I recommend the authors to revise the manuscript again to correct these errors.
Response: We very much appreciate the comment arisen from the reviewer and it has been corrected in Fig. 6b. We are sorry for the grammatical errors, typos, and less-understandable language. We have carefully checked and revised the manuscript and the according changes have been made throughout the manuscript as highlighted with red colors in the revised version.
